# DIMINISHING BATCH NORMALIZATION

## ABSTRACT

In this paper, we propose a generalization of the BN algorithm, diminishing batch normalization (DBN), where we update the BN parameters in a diminishing moving average way. Batch normalization (BN) is very effective in accelerating the convergence of a neural network training phase that it has become a common practice. Our proposed DBN algorithm remains the overall structure of the original BN algorithm while introduces a weighted averaging update to some trainable parameters. We provide an analysis of the convergence of the DBN algorithm that converges to a stationary point with respect to trainable parameters. Our analysis can be easily generalized for original BN algorithm by setting some parameters to constant. To the best knowledge of authors, this analysis is the first of its kind for convergence with Batch Normalization introduced. We analyze a two-layer model with arbitrary activation function. The primary challenge of the analysis is the fact that some parameters are updated by gradient while others are not. The convergence analysis applies to any activation function that satisfies our common assumptions. For the analysis, we also show the sufficient and necessary conditions for the stepsizes and diminishing weights to ensure the convergence. In the numerical experiments, we use more complex models with more layers and ReLU activation. We observe that DBN outperforms the original BN algorithm on Imagenet, MNIST, NI and CIFAR-10 datasets with reasonable complex FNN and CNN models.

## 1 INTRODUCTION

Deep neural networks (DNN) have shown unprecedented success in various applications such as object detection. However, it still takes a long time to train a DNN until it converges. Ioffe & Szegedy identified a critical problem involved in training deep networks, internal covariate shift, and then proposed batch normalization (BN) to decrease this phenomenon. BN addresses this problem by normalizing the distribution of every hidden layer's input. In order to do so, it calculates the pre-activation mean and standard deviation using mini-batch statistics at each iteration of training and uses these estimates to normalize the input to the next layer. The output of a layer is normalized by using the batch statistics, and two new trainable parameters per neuron are introduced that capture the inverse operation. It is now a standard practice Bottou et al. (2016); He et al. (2016). While this approach leads to a significant performance jump, to the best of our knowledge, there is no known theoretical guarantee for the convergence of an algorithm with BN. The difficulty of analyzing the convergence of the BN algorithm comes from the fact that not all of the BN parameters are updated by gradients. Thus, it invalidates most of the classical studies of convergence for gradient methods.

In this paper, we propose a generalization of the BN algorithm, diminishing batch normalization (DBN), where we update the BN parameters in a diminishing moving average way. It essentially means that the BN layer adjusts its output according to all past mini-batches instead of only the current one. It helps to reduce the problem of the original BN that the output of a BN layer on a particular training pattern depends on the other patterns in the current mini-batch, which is pointed out by Bottou et al.. By setting the layer parameter we introduce into DBN to a specific value, we recover the original BN algorithm.

We give a convergence analysis of the algorithm with a two-layer batch-normalized neural network and diminishing stepsizes. We assume two layers (the generalization to multiple layers can be made by using the same approach but substantially complicating the notation) and an arbitrary loss function. The convergence analysis applies to any activation function that follows our common

assumption. The main result shows that under diminishing stepsizes on gradient updates and updates on mini-batch statistics, and standard Lipschitz conditions on loss functions DBN converges to a stationary point. As already pointed out the primary challenge is the fact that some trainable parameters are updated by gradient while others are updated by a minor recalculation.

**Contributions.** The main contribution of this paper is in providing a general convergence guarantee for DBN. Specifically, we make the following contributions.

- In section 4, we show the sufficient and necessary conditions for the stepsizes and diminishing weights to ensure the convergence of BN parameters.

- We show that the algorithm converges to a stationary point under a general nonconvex objective function.

This paper is organized as follows. In Section 2, we review the related works and the development of the BN algorithm. We formally state our model and algorithm in Section 3. We present our main results in Sections 4. In Section 5, we numerically show that the DBN algorithm outperforms the original BN algorithm. Proofs for main steps are collected in the Appendix.

## 2 LITERATURE REVIEW

Before the introduction of BN, it has long been known in the deep learning community that input whitening and decorrelation help to speed up the training process. In fact, Orr & Müller show that preprocessing the data by subtracting the mean, normalizing the variance, and decorrelating the input has various beneficial effects for back-propagation. Krizhevsky et al. propose a method called local response normalization which is inspired by computational neuroscience and acts as a form of lateral inhibition, i.e., the capacity of an excited neuron to reduce the activity of its neighbors. Gülçehre & Bengio propose a standardization layer that bears significant resemblance to batch normalization, except that the two methods are motivated by very different goals and perform different tasks.

Inspired by BN, several new works are taking BN as a basis for further improvements. Layer normalization Ba et al. (2016) is much like the BN except that it uses all of the summed inputs to compute the mean and variance instead of the mini-batch statistics. Besides, unlike BN, layer normalization performs precisely the same computation at training and test times. Normalization propagation that Arpit et al. uses data-independent estimations for the mean and standard deviation in every layer to reduce the internal covariate shift and make the estimation more accurate for the validation phase. Weight normalization also removes the dependencies between the examples in a minibatch so that it can be applied to recurrent models, reinforcement learning or generative models Salimans & Kingma (2016). Cooijmans et al. propose a new way to apply batch normalization to RNN and LSTM models.

Given all these flavors, the original BN method is the most popular technique and for this reason our choice of the analysis. To the best of our knowledge, we are not aware of any prior analysis of BN.

BN has the gradient and non-gradient updates. Thus, nonconvex convergence results do not immediately transfer. Our analysis explicitly considers the workings of BN. However, nonconvex convergence proofs are relevant since some small portions of our analysis rely on known proofs and approaches.

Neural nets are not convex, even if the loss function is convex. For classical convergence results with a nonconvex objective function and diminishing learning rate, we refer to survey papers Bertsekas (2011); Bertsekas & Tsitsiklis (2000); Bottou et al. (2016). Bertsekas & Tsitsiklis provide a convergence result with the deterministic gradient with errors. Bottou et al. provide a convergence result with the stochastic gradient. The classic analyses showing the norm of gradients of the objective function going to zero date back to Grippo (1994); Polyak & Tsypkin (1973); Polyak (1987). For strongly convex objective functions with a diminishing learning rate, we learn the classic convergence results from Bottou et al..

## 3 MODEL AND ALGORITHM

The optimization problem for a network is an objective function consisting of a large number of component functions, that reads:

$$\min \bar{f}(\theta, \lambda) = \sum_{i=1}^{N} f_i(X_i : \theta, \lambda), \tag{1}$$

$$\text{subject to } \theta \in P, \lambda \in Q,$$

where $f_i : \mathbb{R}^{n_1} \times \mathbb{R}^{n_2} \to \mathbb{R}, i = 1, ..., N$, are real-valued functions for any data record $X_i$. Index $i$ associates with data record $X_i$ and target response $y_i$ (hidden behind the dependency of $f$ on $i$) in the training set. Parameters $\theta$ include the common parameters updated by gradients directly associated with the loss function, i.e., behind the part that we have a parametric model, while BN parameters $\lambda$ are introduced by the BN algorithm and not updated by gradient methods but by the mini-batch statistics. We define that the derivative of $f_i$ is always taken with respect to $\theta$:

$$\nabla f_i(X_i : \theta, \lambda) := \nabla_\theta f_i(X_i : \theta, \lambda). \tag{2}$$

The deep network we analyze has 2 fully-connected layers with $D_1$ neurons each. The techniques presented can be extended to more layers with additional notation. Each hidden layer computes $y = a(Wu)$ with activation function $a(\cdot)$ and $u$ is the input vector of the layer. We do not need to include an intercept term since the BN algorithm automatically adjusts for it. BN is applied to the output of the first hidden layer.

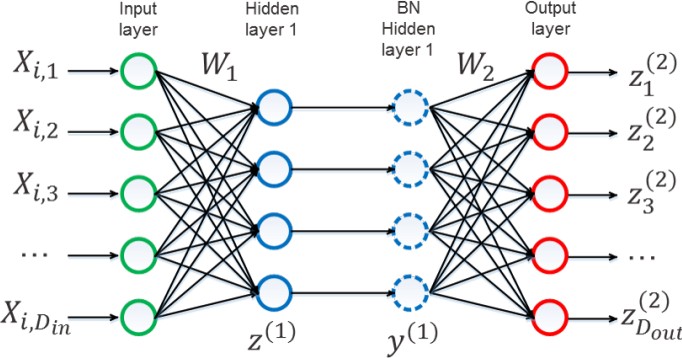

Figure 1: The structure of our batch-normalized network model in the analysis.

We next describe the computation in each layer to show how we obtain the output of the network. The notations introduced here is used in the analysis. Figure 1 shows the full structure of the network. The input data is vector $X$, which is one of $\{X_i\}_{i=1}^{N}$. Vector $\lambda = \left( (\mu_j)_{j=1}^{D}, (\sigma_j)_{j=1}^{D} \right)$ is the set of all BN parameters and vector $\theta = \left( W_1, W_2, (\beta_j^{(1)})_{j=1}^{D}, (\gamma_j^{(1)})_{j=1}^{D} \right)$ is the set of all trainable parameters which are updated by gradients.

Matrices $W_1, W_2$ are the actual model parameters and $\beta, \gamma$ are introduced by BN. The value of $j^{th}$ neuron of the first hidden layer is

$$z_j^{(1)}(X : \theta) = a(W_{1,j,.}X), \tag{3}$$

where $W_{1,j,.}$ denotes the weights of the linear transformations for the $j^{th}$ neuron.

The $j^{th}$ entry of batch-normalized output of the first layer is

$$y_j^{(1)}(X : \theta, \lambda) = \gamma_j^{(1)} \left( \frac{z_j^{(1)}(X : \theta) - \mu_j}{\sigma_j + \epsilon_B} \right) + \beta_j^{(1)},$$

where $\beta_j^{(1)}$ and $\gamma_j^{(1)}$ are trainable parameters updated by gradient and $\mu_j$ and $\sigma_j$ are batch normalization parameters for $z_j^{(1)}$. Trainable parameter $\mu_j$ is the mini-batch mean of $z_j^{(1)}$ and trainable parameter $\sigma_j$ is the mini-batch sample deviation of $z_j^{(1)}$. Constant $\epsilon_B$ keeps the denominator from zero. The output of $j^{th}$ entry of the output layer is:

$$z_j^{(2)}(X:\theta) = a\left(W_{2,j,\cdot}\left[\gamma_j^{(1)}\left(\frac{z_j^{(1)}(X:\theta) - \mu_j}{\sigma_j + \epsilon_B}\right) + \beta_j^{(1)}\right]\right) \tag{4}$$

The objective function for the $i^{th}$ sample is

$$f_i(X_i : \theta, \lambda) = l_i\left(\left(z_j^{(2)}(X_i : \theta, \lambda)\right)_j\right), \tag{5}$$

where $l_i(\cdot)$ is the loss function associated with the target response $y_i$. For sample $i$, we have the following complete expression for the objective function:

$$f_i(X_i : \theta, \lambda) = l_i\left(a(\sum_{j=1}^{D} W_{2,k,j}\left[\gamma_j^{(1)}\frac{a(W_{1,j,\cdot}.X_i - \mu_j)}{\sigma_j + \epsilon_B} + \beta_j^{(1)}\right])_k\right). \tag{6}$$

Function $f_i(X_i : \theta, \lambda)$ is nonconvex with respect to $\theta$ and $\lambda$.

## 3.1 ALGORITHM

Algorithm 1 shows the algorithm studied herein. There are two deviations from the standard BN algorithm, one of them actually being a generalization. We use the full gradient instead of the more popular stochastic gradient (SG) method. It essentially means that each batch contains the entire training set instead of a randomly chosen subset of the training set. An analysis of SG is potential future research. Although the primary motivation for full gradient update is to reduce the burdensome in showing the convergence, the full gradient method is similar to SG in the sense that both of them go through the entire training set, while full gradient goes through it deterministically and the SG goes through it in expectation. Therefore, it is reasonable to speculate that the SG method has similar convergence property as the full algorithm studied herein.

---

**Algorithm 1** DBN: Diminishing Batch-Normalized Network Update Algorithm

---

1: Initialize $\theta \in \mathbb{R}^{n_1}$ and $\lambda \in \mathbb{R}^{n_2}$
2: **for** iteration $m$=1,2,... **do**
3: $\quad \theta^{(m+1)} := \theta^{(m)} - \eta^{(m)} \sum_{i=1}^{N} \nabla f_i(X_i : \theta^{(m)}, \lambda^{(m)})$
4: $\quad$ **for** $j$=1,...,$D_1$ **do**
5: $\quad\quad \mu_j^{(m+1)} := \frac{1}{N}\sum_{i=1}^{N} z_j^{(1)}(X_i : \theta^{(m+1)})$
6: $\quad\quad \sigma_j^{(m+1)} := \sqrt{\frac{1}{N}\sum_{i=1}^{N}\left(z_j^{(1)}(X_i : \theta^{(m+1)}) - \mu_j^{(m+1)}\right)^2}$
7: $\quad \lambda^{(m+1)} := \alpha^{(m+1)}\left((\mu_j^{(m+1)})_{j=1}^{D_1}, (\sigma_j^{(m+1)})_{j=1}^{D_1}\right) + (1 - \alpha^{(m+1)})\lambda^{(m)}$

---

The second difference is that we update the BN parameters $(\theta, \lambda)$ by their moving averages with respect to diminishing $\alpha^{(m)}$. The original BN algorithm can be recovered by setting $\alpha^{(m)} = 1$ for every $m$. After introducing diminishing $\alpha^{(m)}$, $\lambda^{(m)}$ and hence the output of the BN layer is determined by the history of all past data records, instead of those solely in the last batch. Thus, the output of the BN layer becomes more general that better reflects the distribution of the entire dataset. We use two strategies to decide the values of $\alpha^{(m)}$. One is to use a constant smaller than 1 for all $m$, and the other one is to decay the $\alpha^{(m)}$ gradually, such as $\alpha^{(m)} = 1/m$.

In our numerical experiment, we show that Algorithm 1 outperforms the original BN algorithm, where both are based on SG and non-linear activation functions with many layers FNN and CNN models.

## 4 GENERAL CASE

The main purpose of our work is to show that Algorithm 1 converges. In the general case, we focus on the nonconvex objective function.

### 4.1 ASSUMPTIONS

Here are the assumptions we used for the convergence analysis.

**Assumption 1** *(**Lipschitz continuity on** $\theta$ **and** $\lambda$). For every $i$ we have*

$$\|\nabla f_i(X : \tilde{\theta}, \lambda) - \nabla f_i(X : \hat{\theta}, \lambda)\|_2 \leq \bar{L}\|\tilde{\theta} - \hat{\theta}\|_2, \forall \tilde{\theta}, \hat{\theta}, \lambda, X. \tag{7}$$

$$\|\nabla_{W_{1,j,\cdot}} f_i(X : \tilde{\theta}, \lambda) - \nabla_{W_{1,j,\cdot}} f_i(X : \hat{\theta}, \lambda)\|_2$$
$$\leq \bar{L}\|\tilde{W}_{1,j,\cdot} - \hat{W}_{1,j,\cdot}\|_2, \forall \lambda, \tilde{\theta}, \hat{\theta}, X, j \in \{1, ..., D_1\}. \tag{8}$$

$$\|\nabla f_i(X : \theta, \tilde{\lambda}) - \nabla f_i(X : \theta, \hat{\lambda})\|_2 \leq \bar{L}\|\tilde{\lambda} - \hat{\lambda}\|_2,$$
$$\forall \theta, \tilde{\lambda}, \hat{\lambda}, X, j \in \{1, ..., D_1\}. \tag{9}$$

Noted that the Lipschitz constants associated with each of the above inequalities are not necessarily the same. Here $\bar{L}$ is an upper bound for these Lipschitz constants for simplicity.

**Assumption 2** *(**bounded parameters**). Sets $P$ and $Q$ are compact set, where $\theta \in P$ and $\lambda \in Q$. Thus, there exists a constant $M$ that weights $W$ and parameters $\lambda$ are bounded element-wise by this constant $M$.*

$$\|W_1\| \preceq M \text{ and } \|W_2\| \preceq M \text{ and } \|\lambda\| \preceq M.$$

This also implies that the updated $\theta, \lambda$ in Algorithm 1 remain in $P$ and $Q$, respectively.

**Assumption 3** *(**diminishing update on** $\theta$). The stepsizes of $\theta$ update satisfy*

$$\sum_{m=1}^{\infty} \eta^{(m)} = \infty \text{ and } \sum_{m=1}^{\infty} (\eta^{(m)})^2 < \infty. \tag{10}$$

This is a common assumption for diminishing stepsizes in optimization problems.

**Assumption 4** *(**Lipschitz continuity of** $l_i(\cdot)$). Assume the loss functions $l_i(\cdot)$ for every $i$ is continuously differentiable. It implies that there exists $\hat{M}$ such that*

$$\|l_i(x) - l_i(y)\| \leq \hat{M}\|x - y\|, \forall x, y.$$

**Assumption 5** *(**existence of a stationary point**). There exists a stationary point $(\theta^*, \lambda^*)$ such that $\|\nabla \bar{f}(\theta^*, \lambda^*)\| = 0$.*

We note that all these are standard assumptions in convergence proofs. We also stress that Assumption 4 does not directly imply 1. Since we assume that $P$ and $Q$ are compact, then Assumptions 1, 4 and 5 hold for many standard loss function such as softmax and MSE.

**Assumption 6** *(**Lipschitz at activation function**). The activation function $a(\cdot)$ is Lipschitz with constant $k$:*

$$|a(x)| \leq k\|x\| \tag{11}$$

Since for all activation function there is $a(0) = 0$, the condition is equivalent to $|a(x) - a(0)| \leq k\|x - 0\|$. We note that this assumption works for many popular choices of activation functions, such as ReLU and LeakyReLu.

## 4.2 CONVERGENCE ANALYSIS

We first have the following lemma specifying sufficient conditions for $\lambda$ to converge. Proofs for main steps are given in the Appendix.

**Theorem 7** *Under Assumptions 1, 2, 3 and 6, if $\{\alpha^{(m)}\}$ satisfies*

$$\sum_{m=1}^{\infty} \alpha^{(m)} < \infty \text{ and } \sum_{m=1}^{\infty} \sum_{n=1}^{m} \alpha^{(m)} \eta^{(n)} < \infty,$$

*then sequence $\{\lambda^{(m)}\}$ converges to $\bar{\lambda}$.*

We give a discussion of the above conditions for $\alpha^{(m)}$ and $\eta^{(m)}$ at the end of this section. With the help of Theorem 7, we can show the following convergence result.

**Lemma 8** *Under Assumptions 4, 5 and the assumptions of Theorem 7, when*

$$\sum_{m=1}^{\infty} \sum_{i=m}^{\infty} \sum_{n=1}^{i} \alpha^{(i)} \eta^{(n)} < \infty \quad and \quad \sum_{m=1}^{\infty} \sum_{n=m}^{\infty} \alpha^{(n)} < \infty, \tag{12}$$

*we have*

$$\limsup_{M \to \infty} \sum_{m=1}^{M} \eta^{(m)} \|\nabla \bar{f}(\theta^{(m)}, \bar{\lambda})\|_2^2 < \infty. \tag{13}$$

This result is similar to the classical convergence rate analysis for the non-convex objective function with diminishing stepsizes, which can be found in Bottou et al. (2016).

**Lemma 9** *Under the assumptions of Lemma 8, we have*

$$\liminf_{m \to \infty} \|\nabla \bar{f}(\theta^{(m)}, \bar{\lambda})\|_2^2 = 0. \tag{14}$$

This theorem states that for the full gradient method with diminishing stepsizes the gradient norms cannot stay bounded away from zero. The following result characterizes more precisely the convergence property of Algorithm 1.

**Lemma 10** *Under the assumptions stated in Lemma 8, we have*

$$\lim_{m \to \infty} \|\nabla \bar{f}(\theta^{(m)}, \bar{\lambda})\|_2^2 = 0. \tag{15}$$

Our main result is listed next.

**Theorem 11** *Under the assumptions stated in Lemma 8, we have*

$$\lim_{m \to \infty} \|\nabla \bar{f}(\theta^{(m)}, \lambda^{(m)})\|_2^2 = 0. \tag{16}$$

We cannot show that $\{\theta^{(m)}\}$'s converges (standard convergence proofs are also unable to show such a stronger statement). For this reason, Theorem 11 does not immediately follow from Lemma 10 together with Theorem 7. The statement of Theorem 11 would easily follow from Lemma 10 if the convergence of $\{\theta^{(m)}\}$ is established and the gradient being continuous.

Considering the cases $\eta^{(m)} = O(\frac{1}{m^k})$ and $\alpha^{(m)} = O(\frac{1}{m^h})$. We show in the Appendix that the set of sufficient and necessary conditions to satisfy the assumptions of Theorem 7 are $h > 1$ and $k \geq 1$. The set of sufficient and necessary conditions to satisfy the assumptions of Lemma 8 are $h > 2$ and $k \geq 1$. For example, we can pick $\eta^{(m)} = O(\frac{1}{m})$ and $\alpha^{(m)} = O(\frac{1}{m^{2.001}})$ to achieve the above convergence result in Theorem 11.

## 5 COMPUTATIONAL EXPERIMENTS

We conduct the computational experiments with Theano and Lasagne on a Linux server with a Nvidia Titan-X GPU. We use MNIST LeCun et al. (1998), CIFAR-10 Krizhevsky & Hinton (2009) and Network Intrusion (NI) kdd (1999) datasets to compare the performance between DBN and the original BN algorithm. For the MNIST dataset, we use a four-layer fully connected FNN ($784 \times 300 \times 300 \times 10$) with the ReLU activation function and for the NI dataset, we use a four-layer fully connected FNN ($784 \times 50 \times 50 \times 10$) with the ReLU activation function. For the CIFAR-10 dataset, we use a reasonably complex CNN network that has a structure of (Conv-Conv-MaxPool-Dropout-Conv-Conv-MaxPool-Dropout-FC-Dropout-FC), where all four convolution layers and the first fully connected layers are batch normalized. We use the softmax loss function and $l_2$ regularization with for all three models. All the trainable parameters are randomly initialized before training. For all 3 datasets, we use the standard epoch/minibatch setting with the minibatch size of $100$, i.e., we do not compute the full gradient and the statistics are over the minibatch. We use AdaGrad Duchi, John and Hazan, Elad and Singer (2011) to update the learning rates $\eta^{(m)}$ for trainable parameters, starting from $0.01$.

We use two different strategies to decide the values of $\alpha^{(m)}$ in DBN: constant values of $\alpha^{(m)}$ and diminishing $\alpha^{(m)}$ where $\alpha^{(m)} = 1/m$ and $\alpha^{(m)} = 1/m^2$. We test the choices of constant $\alpha^{(m)} \in \{1, 0.75, 0.5, 0.25, 0.1, 0.01, 0.001, 0\}$.

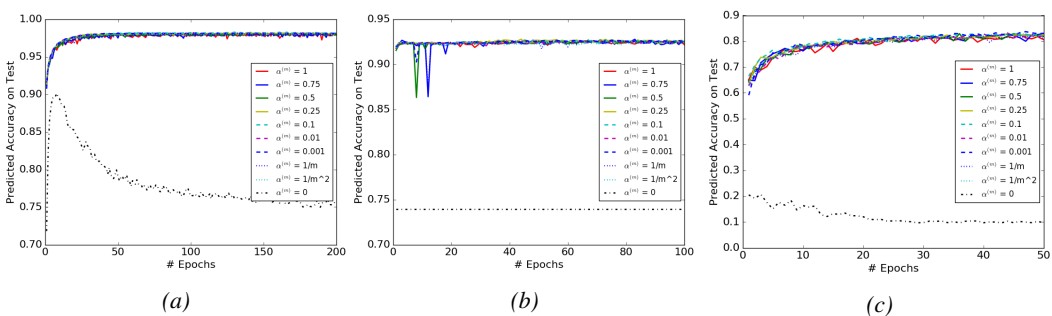

Figure 2: Comparison of predicted accuracy on test datasets for different choices of $\alpha^{(m)}$. From left to right are FNN on MNIST, FNN on NI and CNN on CIFAR-10.

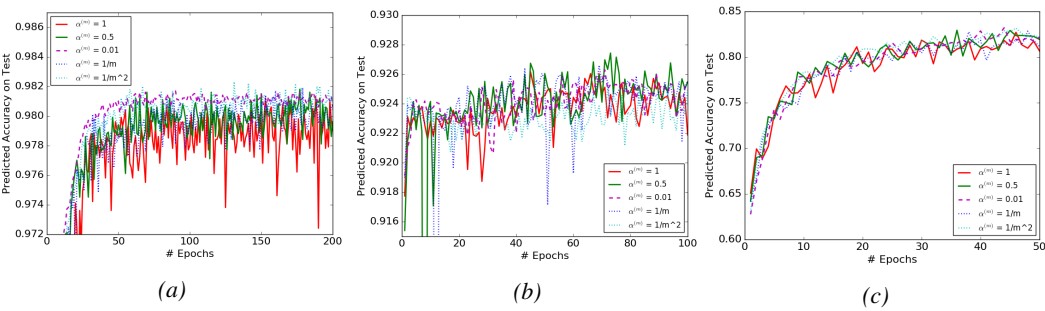

Figure 3: Comparison of predicted accuracy on test datasets for the most efficient choices of $\alpha^{(m)}$. From left to right are FNN on MNIST, FNN on NI and CNN on CIFAR-10.

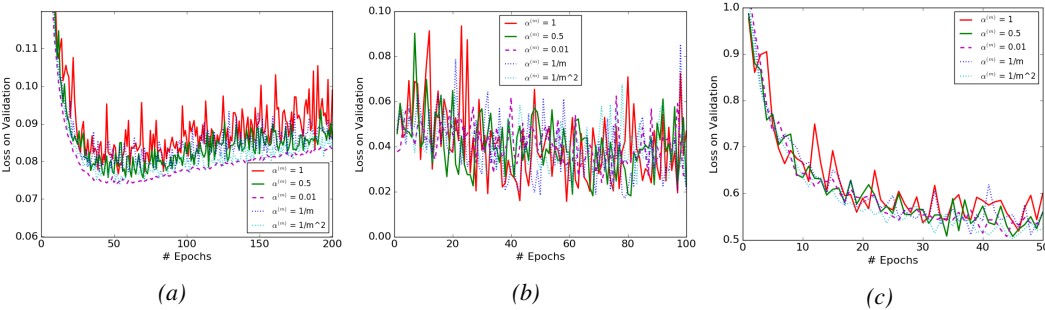

*Figure 4: Comparison of the convergence of the loss function value on the validation set for different choices of $\alpha^{(m)}$. From left to right are FNN on MNIST, FNN on NI and CNN on CIFAR-10.*

We test all the choices of $\alpha^{(m)}$ with the performances presented in Figure 2. Figure 2 shows that all the non-zero choices of $\alpha^{(m)}$ converge properly. The algorithms converge without much difference even when $\alpha^{(m)}$ in DBN is very small, e.g., $1/m^2$. However, if we select $\alpha^{(m)} = 0$, the algorithm is erratic. Besides, we observe that all the non-zero choices of $\alpha^{(m)}$ converge at a similar rate. The fact that DBN keeps the batch normalization layer stable with a very small $\alpha^{(m)}$ suggests that the BN parameters do not have to be depended on the latest minibatch, i.e., the original BN.

We compare a selected set of the most efficient choices of $\alpha^{(m)}$ in Figures 3 and 4. They show that DBN with $\alpha^{(m)} < 1$ is more stable than the original BN algorithm. The variances with respect to epochs of the DBN algorithm are smaller than those of the original BN algorithms in each figure.

*Table 1: Best results for different choices of $\alpha^{(m)}$ on each dataset, showing the top three with a heat map.*

| | Test Error | | |
|---|---|---|---|
| Model | MNIST | NI | CIFAR-10 |
| $\alpha^{(m)} = 1$ | 2.70% | 7.69% | 17.31% |
| $\alpha^{(m)} = 0.75$ | 1.91% | 7.37% | 17.03% |
| $\alpha^{(m)} = 0.5$ | **1.84%** | 7.46% | 17.11% |
| $\alpha^{(m)} = 0.25$ | 1.91% | **7.24%** | 17.00% |
| $\alpha^{(m)} = 0.1$ | 1.90% | 7.36% | 17.10% |
| $\alpha^{(m)} = 0.01$ | 1.94% | 7.47% | 16.82% |
| $\alpha^{(m)} = 0.001$ | 1.95% | 7.43% | **16.28%** |
| $\alpha^{(m)} = 1/m$ | 2.10% | 7.45% | 17.26% |
| $\alpha^{(m)} = 1/m^2$ | 2.00% | 7.59% | 17.23% |
| $\alpha^{(m)} = 0$ | 24.27% | 26.09% | 79.34% |

Table 1 shows the best result obtained from each choice of $\alpha^{(m)}$. Most importantly, it suggests that the choices of $\alpha^{(m)} = 1/m$ and $1/m^2$ perform better than the original BN algorithm. Besides, all the constant less-than-one choices of $\alpha^{(m)}$ perform better than the original BN, showing the importance of considering the mini-batch history for the update of the BN parameters. The BN algorithm in each figure converges to similar error rates on test datasets with different choices of $\alpha^{(m)}$ except for the $\alpha^{(m)} = 0$ case. Among all the models we tested, $\alpha^{(m)} = 0.25$ is the only one that performs top 3 for all three datasets, thus the most robust choice.

To summarize, our numerical experiments show that the DBN algorithm outperforms the original BN algorithm on the MNIST, NI and CIFAT-10 datasets with typical deep FNN and CNN models.

**Future Directions.** On the analytical side, we believe an extension to more than 2 layers is doable with significant augmentations of the notation. A stochastic gradient version is likely to be much more challenging to analyze. A second open question concerns more general activation functions. It would be interesting to analyze other activation functions, such as Sigmoid, that do not apply to our current assumptions.

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

## 6 APPENDIX: PROOFS

### 6.1 PRELIMINARY RESULTS

The following proofs are shortened to corporate with AAAI submission page limit.

**Proposition 12** *There exists a constant M such that, for any $\theta$ and fixed $\lambda$, we have*
$$\|\nabla \bar{f}(\theta, \lambda)\|_2^2 \leq M. \tag{17}$$

*Proof.* By Assumption 5, we know there exists $(\theta^*, \lambda^*)$ such that $\|\nabla \bar{f}(\theta^*, \lambda^*)\|_2 = 0$. Then we have

$$
\begin{aligned}
&\|\nabla \bar{f}(\theta, \lambda)\|_2 \\
=&\|\nabla \bar{f}(\theta, \lambda)\|_2 - \|\nabla \bar{f}(\theta, \lambda^*)\|_2 + \|\nabla \bar{f}(\theta, \lambda^*)\|_2 - \|\nabla \bar{f}(\theta^*, \lambda^*)\|_2 \\
\leq&\|\nabla \bar{f}(\theta, \lambda) - \nabla \bar{f}(\theta, \lambda^*)\|_2 + \|\nabla \bar{f}(\theta, \lambda^*) - \nabla \bar{f}(\theta^*, \lambda^*)\|_2 \\
\leq& \sum_{i=1}^{N} \|\nabla f_i(X_i : \theta, \lambda) - \nabla f_i(X_i : \theta, \lambda^*)\|_2 \\
&+ \sum_{i=1}^{N} \|\nabla f_i(X_i : \theta, \lambda^*) - \nabla f_i(X_i : \theta^*, \lambda^*)\|_2 \\
\leq& N\bar{L}(\|\lambda - \lambda^*\|_2 + \|\theta - \theta^*\|_2),
\end{aligned}
\tag{18}
$$

where the last inequality is by Assumption 1. We then have
$$\|\nabla \bar{f}(\theta, \lambda)\|_2^2 \leq N^2 \bar{L}^2 (\|\lambda - \lambda^*\|_2 + \|\theta - \theta^*\|_2)^2 \leq M, \tag{19}$$
because sets P and Q are compact by Assumption 2. □

**Proposition 13** *We have*
$$f_i(X : \tilde{\theta}, \lambda) \leq f_i(X : \hat{\theta}, \lambda) + \nabla f_i(X : \hat{\theta}, \lambda)^T (\tilde{\theta} - \hat{\theta}) + \frac{1}{2}\bar{L}\|\tilde{\theta} - \hat{\theta}\|_2^2, \forall \tilde{\theta}, \hat{\theta}, X. \tag{20}$$

*Proof.* This is a known result of the Lipschitz-continuous condition that can be found in Bottou et al. (2016). We have this result together with Assumption 1.

### 6.2 PROOF OF THEOREM 7

**Lemma 14** *When $\sum_{m=1}^{\infty} \alpha^{(m)} < \infty$ and $\sum_{m=1}^{\infty} \sum_{n=1}^{m} \alpha^{(m)} \eta^{(n)} < \infty$,*
$\tilde{\mu}_j^{(m)} := \dfrac{\mu_j^{(m)}}{(1 - \alpha^{(1)})(1 - \alpha^{(2)})...(1 - \alpha^{(m)})}$ *is a Cauchy series.*

*Proof.* By Algorithm 1, we have
$$\mu_j^{(m)} = \alpha^{(m)} \frac{1}{N} \sum_{i=1}^{N} a(W_{1,j,\cdot}^{(m)} X_i) + (1 - \alpha^{(m)})\mu_j^{(m-1)}. \tag{21}$$

We define $\tilde{\alpha}^{(m)} := \dfrac{\alpha^{(m)}}{(1 - \alpha^{(1)})(1 - \alpha^{(2)})...(1 - \alpha^{(m)})}$ and $\Delta W_{1,j,\cdot}^{(m)} := W_{1,j,\cdot}^{(m)} - W_{1,j,\cdot}^{(m-1)}$. After dividing equation 21 by $(1 - \alpha^{(1)})(1 - \alpha^{(2)})...(1 - \alpha^{(m)})$, we obtain

$$\tilde{\mu}_j^{(m)} = \tilde{\alpha}^{(m)} \frac{1}{N} \sum_{i=1}^{N} a(W_{1,j,\cdot}^{(m)} X_i) + \tilde{\mu}_j^{(m-1)}. \tag{22}$$

Then we have

$$|\tilde{\mu}_j^{(m)} - \tilde{\mu}_j^{(m-1)}| \leq \tilde{\alpha}^{(m)} |k| \frac{1}{N} \sum_{i=1}^{N} |\sum_{n=1}^{m} \Delta W_{1,j,\cdot}^{(n)} X_i|$$

$$= \tilde{\alpha}^{(m)} |k| \frac{1}{N} \sum_{i=1}^{N} \left| \sum_{n=1}^{m} \left( \eta^{(n)} \sum_{l=1}^{N} \nabla_{W_{1,j,\cdot}} f_l(X_l : \theta^{(n)}, \lambda^{(n)}) \right) \cdot X_i \right|$$

$$= \tilde{\alpha}^{(m)} |k| \frac{1}{N} \sum_{i=1}^{N} \sum_{n=1}^{m} \left( \eta^{(n)} \left| \left( \sum_{l=1}^{N} \nabla_{W_{1,j,\cdot}} f_l(X_l : \theta^{(n)}, \lambda^{(n)}) \right) \cdot X_i \right| \right)$$

$$\leq \tilde{\alpha}^{(m)} |k| \frac{1}{N} \sum_{i=1}^{N} \sum_{n=1}^{m} \left( \eta^{(n)} \| \sum_{l=1}^{N} \nabla_{W_{1,j,\cdot}} f_l(X_l : \theta^{(n)}, \lambda^{(n)}) \| \cdot \|X_i\| \right) \tag{23}$$

$$\leq \tilde{\alpha}^{(m)} |k| \sum_{i=1}^{N} \sum_{n=1}^{m} \eta^{(n)} \left( \bar{L} \cdot (\|W_{1,j,\cdot}^{(n)} - W_{1,j,\cdot}^*\|_2 + \|\lambda_{j,\cdot}^{(n)} - \lambda_{j,\cdot}^*\|_2) \cdot \|X_i\|_2 \right) \tag{24}$$

$$\leq \tilde{\alpha}^{(m)} \sum_{n=1}^{m} \left( \eta^{(n)} \right) |k| \sum_{i=1}^{N} \left( 2\bar{L} M \|X_i\|_2 \right) \tag{25}$$

$$\leq \tilde{\alpha}^{(m)} \sum_{n=1}^{m} \eta^{(n)} \tilde{M}_{\bar{L},M}. \tag{26}$$

Equation equation 6.2 is due to $W_{1,i,j}^{(m)} = \sum_{n=1}^{m} \Delta W_{1,i,j}^{(n)}$.

Therefore,

$$|\tilde{\mu}_j^{(p)} - \tilde{\mu}_j^{(q)}| \leq \tilde{M}_{\bar{L},M} \cdot \sum_{m=p}^{q} \sum_{n=1}^{m} \tilde{\alpha}^{(m)} \eta^{(n)}. \tag{27}$$

It remains to show that

$$\sum_{m=1}^{\infty} \alpha^{(m)} < \infty, \tag{28}$$

$$\sum_{m=1}^{\infty} \sum_{n=1}^{m} \alpha^{(m)} \eta^{(n)} < \infty, \tag{29}$$

implies the convergence of $\{\tilde{\mu}^{(m)}\}$. By (28), we have $\Pi_{m=1}^{\infty}(1 - \alpha^{(m)}) > 0$, since $\ln(\Pi_{m=1}^{\infty}(1 - \alpha^{(m)})) > \sum_{m=1}^{\infty} -\alpha^{(m)} > -\infty$.

It is also easy to show that there exists $C$ and $M_c$ such that for all $m \geq M_c$, we have

$$(1 - \alpha^{(1)})(1 - \alpha^{(2)}) \ldots (1 - \alpha^{(m)}) \geq C. \tag{30}$$

Therefore, $\lim_{m \to \infty} (1 - \alpha^{(1)})(1 - \alpha^{(2)}) \ldots (1 - \alpha^{(m)}) \geq C$.

Thus the following holds:

$$\tilde{\alpha}^{(m)} \leq \frac{1}{C} \alpha^{(m)} \tag{31}$$

and

$$\sum_{m=p}^{q} \sum_{n=1}^{m} \tilde{\alpha}^{(m)} \eta^{(n)} \leq \frac{1}{C} \sum_{m=p}^{q} \sum_{n=1}^{m} \alpha^{(m)} \eta^{(n)}. \tag{32}$$

From equation 29 and equation 32 it follows that the sequence $\{\tilde{\mu}_j^{(m)}\}$ is a Cauchy series. $\quad\square$

**Lemma 15** *Since $\{\tilde{\mu}_j^{(m)}\}$ is a Cauchy series, $\{\mu_j^{(m)}\}$ is a Cauchy series.*

*Proof.* We know that $\mu_j^{(m)} = \tilde{\mu}_j^{(m)}(1 - \alpha^{(1)}) \ldots (1 - \alpha^{(m)})$. Since $\lim_{m \to \infty} \tilde{\mu}_j^{(m)} \to \tilde{\mu}_j$ and $\lim_{m \to \infty} (1 - \alpha^{(1)}) \ldots (1 - \alpha^{(m)}) \to \tilde{C}$, we have $\lim_{m \to \infty} \mu_j^{(m)} \to \tilde{\mu}_j \cdot \tilde{C}$. Thus $\mu_j^{(m)}$ is a Cauchy series. $\quad\square$

**Lemma 16** *If $\sum_{m=1}^{\infty} \alpha^{(m)} < \infty$ and $\sum_{m=1}^{\infty} \sum_{n=1}^{m} \alpha^{(m)} \eta^{(n)} < \infty$, $\{\sigma_j^{(m)}\}$ is a Cauchy series.*

*Proof.* We define $\sigma_j^{(m)} := \tilde{\sigma}_j^{(m)}(1 - \alpha^{(1)}) \ldots (1 - \alpha^{(m)})$. Then we have

$$|\tilde{\sigma}_j^{(m+1)} - \tilde{\sigma}_j^{(m)}| = \tilde{\alpha}^{(m)} \sqrt{\frac{1}{N} \sum_{i=1}^{N} \left( a(W_{1,j,\cdot}^{(m)} X_i) - \mu_j^{(m)} \right)^2}$$

$$= \tilde{\alpha}^{(m)} \frac{|k|}{\sqrt{N}} \sqrt{\sum_{i=1}^{N} \left( \frac{a(W_{1,j,\cdot}^{(m)} X_i)}{k} - \frac{\mu_j^{(m)}}{k} \right)^2}. \tag{33}$$

Since $\{\mu_j^{(m)}\}$ is convergent, there exists $c_1, c_2$ and $N_1$ such that for any $m > N_1$, $-\infty < c_1 < \mu_j^{(m)} < c_2 < \infty$. For any $\bar{C} \in \left\{ \frac{c_1}{k}, \frac{c_2}{k} \right\}$, we have

$$|\tilde{\sigma}_j^{(m+1)} - \tilde{\sigma}_j^{(m)}| \leq \tilde{\alpha}^{(m)} \frac{|k|}{\sqrt{N}} \cdot \sqrt{\sum_{i=1}^{N} \left( \frac{a(W_{1,j,\cdot}^{(m)} X_i)}{k} - \bar{C} \right)^2} \tag{34}$$

$$\leq \tilde{\alpha}^{(m)} \frac{|k|}{\sqrt{N}} \cdot \sqrt{\sum_{i=1}^{N} \left( |\frac{a(W_{1,j,\cdot}^{(m)} X_i)}{k}| + |\bar{C}| \right)^2} \tag{35}$$

$$\leq \tilde{\alpha}^{(m)} \frac{|k|}{\sqrt{N}} \cdot \sqrt{\sum_{i=1}^{N} \left( \sum_{n=1}^{m} \eta^{(n)} \left( 2N\bar{L}M \|X_i\|_2 \right) + |\bar{C}| \right)^2} \tag{36}$$

$$\leq \tilde{\alpha}^{(m)} \frac{|k|}{\sqrt{N}} \cdot \sqrt{N \cdot \left( \tilde{M}_{\bar{L},M} \sum_{n=1}^{m} \eta^{(n)} + |\bar{C}| \right)^2} \tag{37}$$

$$= \tilde{\alpha}^{(m)} |k| \cdot \left( \tilde{M}_{\bar{L},M} \sum_{n=1}^{m} \eta^{(n)} + |\bar{C}| \right). \tag{38}$$

Inequality equation 35 is by the following fact:

$$\sqrt{\sum_{i=1}^{n} (a_i - c)^2} \leq \sqrt{\sum_{i=1}^{n} (|a_i| + |c|)^2}, \tag{39}$$

where $b$ and $a_i$ for every $i$ are arbitrary real scalars. Besides, equation 39 is due to $-2a_i c \leq \max\{-2|a_i|c, 2|a_i|c\}$.

Inequality equation 36 follow from the square function being increasing for nonnegative numbers. Besides these facts, equation 36 is also by the same techniques we used in equation 23-equation 25 where we bound the derivatives with the Lipschitz continuity in the following inequality:

$$\|\sum_{l=1}^{N} \nabla_{W_{1,j,\cdot}} f_l(X_l : \theta^{(n)}, \lambda^{(n)})\| \leq 2N\bar{L}M. \tag{40}$$

Inequality equation 37 is by collecting the bounded terms into a single bound $\tilde{M}_{\bar{L},M}$. Therefore,

$$|\tilde{\sigma}_j^{(q)} - \tilde{\sigma}_j^{(p)}| \leq \sum_{m=p}^{q-1} \tilde{\alpha}^{(m)} |k| \cdot \left( \tilde{M}_{\bar{L},M} \sum_{n=1}^{m} \eta^{(n)} + |\bar{C}| \right). \tag{41}$$

Using the similar methods in deriving equation 28 and equation 29, it can be seen that a set of sufficient conditions ensuring the convergence for $\{\tilde{\sigma}_j^{(m)}\}$ is: $\sum_{m=1}^{\infty} \alpha^{(m)} < \infty$ and $\sum_{m=1}^{\infty} \sum_{n=1}^{m} \alpha^{(m)} \eta^{(n)} < \infty$.

Therefore, the convergence conditions for $\{\sigma_j^{(m)}\}$ are the same as for $\{\mu_j^{(m)}\}$. $\qquad \square$

It is clear that these lemmas establish the proof of Theorem 7.

### 6.3 CONSEQUENCES OF THEOREM 7

**Proposition 17** *Under the assumptions of Theorem 7, we have* $|\lambda^{(m)} - \bar{\lambda}|_\infty \leq a_m$, *where*

$$a_m = M_1 \sum_{i=m}^{\infty} \sum_{j=1}^{i} \alpha^{(i)} \eta^{(j)} + M_2 \sum_{i=m}^{\infty} \alpha^{(i)}. \tag{42}$$

$M_1$ *and* $M_2$ *are constants.*

*Proof.* For the upper bound of $\sigma_j^{(m)}$, by equation 38, we have

$$|\tilde{\sigma}_j^{(q)} - \tilde{\sigma}_j^{(p)}| \leq \sum_{m=p}^{q-1} \tilde{\alpha}^{(m)} |k| \left( \tilde{M}_{\bar{L},M} \sum_{n=1}^{m} \eta^{(n)} + |\bar{C}| \right).$$

We define $\tilde{\sigma}_j := \dfrac{\bar{\sigma}_j}{(1-\alpha^{(1)})...(1-\alpha^{(u)})...}$. Therefore,

$$\begin{aligned}
|\tilde{\sigma}_j - \tilde{\sigma}_j^{(m)}| &\leq \sum_{i=m}^{\infty} \tilde{\alpha}^{(i)} |k| \left( \tilde{M}_{\bar{L},M} \sum_{j=1}^{i} \eta^{(j)} + |\bar{C}| \right) \\
&\leq \frac{|k|}{C} \sum_{i=m}^{\infty} \alpha^{(i)} \left( \tilde{M}_{\bar{L},M} \sum_{j=1}^{i} \eta^{(j)} + |\bar{C}| \right).
\end{aligned} \tag{43}$$

The first inequality comes by substituting $p$ by $m$ and by taking lim as $q \to \infty$ in equation 41. The second inequality comes from equation 30. We then obtain,

$$\begin{aligned}
\left| \sigma_j^{(m)} - \bar{\sigma}_j \right| &\leq \left| \tilde{\sigma}_j^{(m)} - \tilde{\sigma}_j^{(\infty)} \right| + \left| \frac{\bar{\sigma}_j}{(1-\alpha^{(1)})...(1-\alpha^{(m)})} - \tilde{\sigma}_j^{(\infty)} \right| \\
&= \left| \tilde{\sigma}_j^{(m)} - \tilde{\sigma}_j^{(\infty)} \right| + \left| \frac{\bar{\sigma}_j}{(1-\alpha^{(1)})...(1-\alpha^{(m)})} - \frac{\bar{\sigma}_j}{(1-\alpha^{(1)})...(1-\alpha^{(u)})...} \right| \\
&= \left| \tilde{\sigma}_j^{(m)} - \tilde{\sigma}_j^{(\infty)} \right| + \bar{\sigma}_j \left| \frac{(1-\alpha^{(m+1)})...(1-\alpha^{(u)})... - 1}{(1-\alpha^{(1)})...(1-\alpha^{(u)})...} \right| \\
&\leq \left| \tilde{\sigma}_j^{(m)} - \tilde{\sigma}_j^{(\infty)} \right| + \frac{\bar{\sigma}_j}{C} |1 - (1-\alpha^{(m+1)})...(1-\alpha^{(u)})...| \\
&\leq \left| \tilde{\sigma}_j^{(m)} - \tilde{\sigma}_j^{(\infty)} \right| + \frac{\bar{\sigma}_j}{C} \sum_{n=m+1}^{\infty} \alpha^{(n)}.
\end{aligned} \tag{44}$$

The second inequality is by $(1-\alpha^{(1)})...(1-\alpha^{(m)}) < 1$, the third inequality is by equation 30 and the last inequality can be easily seen by induction. By equation 44, we obtain

$$|\bar{\sigma}_j - \sigma_j^{(m)}| = \lim_{M \to \infty} |\sigma_j^{(M)} - \sigma_j^{(m)}| \leq |\tilde{\sigma}_j - \tilde{\sigma}_j^{(m)}| + \frac{\bar{\sigma}_j}{C} \sum_{n=m+1}^{\infty} \alpha^{(n)}. \tag{45}$$

Therefore, we have

$$\begin{aligned}
|\bar{\sigma}_j - \sigma_j^{(m)}| &\leq |\tilde{\sigma}_j - \tilde{\sigma}_j^{(m)}| + \frac{\bar{\sigma}_j}{C} \sum_{n=m+1}^{\infty} \alpha^{(n)} \\
&\leq \sum_{i=m}^{\infty} \tilde{\alpha}^{(i)} |k| \cdot \left( \tilde{M}_{\bar{L},M} \sum_{j=1}^{i} \eta^{(j)} + |\bar{C}| \right) + \frac{\bar{\sigma}_j}{C} \sum_{i=m+1}^{\infty} \alpha^{(i)} \\
&\leq \sum_{i=m}^{\infty} \frac{1}{C} \alpha^{(i)} |k| \cdot \left( \tilde{M}_{\bar{L},M} \sum_{j=1}^{i} \eta^{(j)} + |\bar{C}| \right) + \frac{\bar{\sigma}_j}{C} \sum_{i=m+1}^{\infty} \alpha^{(i)} \\
&\leq \frac{\tilde{M}_{\bar{L},M} |k|}{C} \sum_{i=m}^{\infty} \sum_{j=1}^{i} \alpha^{(i)} \eta^{(j)} + \left( \frac{\bar{\sigma}_j}{C} + \frac{|k||\bar{C}|}{C} \right) \sum_{i=m}^{\infty} \alpha^{(i)}.
\end{aligned} \tag{46}$$

The first inequality is by equation 45, the second inequality is by equation 41, the third inequality is by equation 31 and the fourth inequality is by adding the nonnegative term $\dfrac{\bar{\sigma}_j}{C} \alpha^{(m)}$ to the right-hand side.

For the upper bound of $\mu_j^{(m)}$, we have

$$\left| \mu_j^{(m)} - \bar{\mu}_j \right| \leq \left| \tilde{\mu}^{(m)} - \tilde{\mu}^{(\infty)} \right| + \left| \frac{\bar{\mu}_j}{(1-\alpha^{(1)})...(1-\alpha^{(m)})} - \tilde{\mu}^{(\infty)} \right|. \tag{47}$$

Let us define $A_m := \left| \tilde{\mu}^{(m)} - \tilde{\mu}^{(\infty)} \right|$ and $B_m := \left| \dfrac{\bar{\mu}_j}{(1-\alpha^{(1)})...(1-\alpha^{(m)})} - \tilde{\mu}^{(\infty)} \right|$. Recall from Theorem 7 that $\{\mu_j^{(m)}\}$ is a Cauchy series, by equation 27, $|\tilde{\mu}_j^{(p)} - \tilde{\mu}_j^{(q)}| \leq \bar{M}_{\bar{L},M} \cdot \sum_{m=p}^{q} \sum_{n=1}^{m} \alpha^{(m)} \eta^{(n)}$. Therefore, the first term in equation 47 is bounded by

$$|\tilde{\mu}_j^{(m)} - \tilde{\mu}_j^{\infty}| \leq \tilde{M}_{\bar{L},M} \cdot \sum_{i=m}^{\infty} \sum_{n=1}^{i} \alpha^{(i)} \eta^{(n)} < \infty. \tag{48}$$

For the second term in equation 47, recall that $C := (1 - \alpha^{(1)})...(1 - \alpha^{(u)})....$ Then we have

$$C \cdot \left| \frac{\bar{\mu}_j}{(1 - \alpha^{(1)})...(1 - \alpha^{(m)})} - \tilde{\mu}^{(\infty)} \right| \leq \bar{\mu}_j \sum_{i=m+1}^{\infty} \alpha^{(i)}, \quad \text{where the inequality can be easily seen by in-}$$

duction. Therefore, the second term in equation 47 is bounded by

$$\left| \frac{\bar{\mu}_j}{(1 - \alpha^{(1)})...(1 - \alpha^{(m)})} - \tilde{\mu}^{(\infty)} \right| \leq \frac{\bar{\mu}_j}{C} \sum_{i=m+1}^{\infty} \alpha^{(i)}. \tag{49}$$

From these we obtain

$$\left| \mu_j^{(m)} - \bar{\mu}_j \right| \leq \tilde{M}_{\bar{L},M} \sum_{i=m}^{\infty} \sum_{n=1}^{i} \alpha^{(i)} \eta^{(n)} + \frac{\bar{\mu}_j}{C} \sum_{i=m+1}^{\infty} \alpha^{(i)}. \tag{50}$$

The first inequality is by equation 47 and the second inequality is by equation 48 and equation 49. Combining equation 46 and equation 50, we have that

$$|\lambda^{(m)} - \bar{\lambda}|_\infty \leq M_1 \sum_{i=m}^{\infty} \sum_{j=1}^{i} \alpha^{(i)} \eta^{(j)} + M_2 \sum_{i=m}^{\infty} \alpha^{(i)},$$

where $M_1$ and $M_2$ are constants defined as $M_1 = \max(\frac{\tilde{M}_{\bar{L},M}|k|}{C}, \bar{M}_{\bar{L},M})$ and $M_2 = \max(\frac{\bar{\sigma}_j + |k||\bar{C}|}{C}, \frac{\bar{\mu}_j}{C})$. $\qquad \square$

**Proposition 18** *Under the assumptions of Theorem 7,*
$$-\nabla \bar{f}(\theta^{(m)}, \bar{\lambda})^T \cdot \nabla \bar{f}(\theta^{(m)}, \lambda^{(m)}) \leq -\|\nabla \bar{f}(\theta^{(m)}, \bar{\lambda})\|^2 + \bar{L}M\sqrt{n_2}a_m,$$
*where $a_m$ is defined in Proposition 17.*

*Proof.* For simplicity of the proof, let us define $x^{(m)} := \nabla \bar{f}(\theta^{(m)}, \bar{\lambda})$, $y^{(m)} := \nabla \bar{f}(\theta^{(m)}, \lambda^{(m)})$. We have
$$|x^{(m)} - y^{(m)}|_\infty \leq \bar{L}\sqrt{n_2}\|\lambda^{(m)} - \bar{\lambda}\|_\infty \leq \bar{L}\sqrt{n_2}a_m, \tag{51}$$
where $\sqrt{n_2}$ is the dimension of $\lambda$. The second inequality is by Assumption 1 and the fourth inequality is by Proposition 17. Inequality equation 51 implies that for all $m$ and $i$, we have $|x_i^{(m)} - y_i^{(m)}| \leq \bar{L}\sqrt{n_2}a_m$.

It remains to show
$$-\sum_i y_i^{(m)} x_i^{(m)} \leq -\sum_i x_i^{(m)2} + \bar{L}M\sqrt{n_2}a_m, \forall i, m. \tag{52}$$

This is established by the following four cases.

1) If $x_i^{(m)} \geq 0, x_i^{(m)} - y_i^{(m)} \geq 0$, then $x_i^{(m)} \leq \bar{L}\sqrt{n_2}a_m + y_i^{(m)}$. Thus $-x_i^{(m)} y_i^{(m)} \leq -x_i^{(m)2} + \bar{L}M\sqrt{n_2}a_m$ by Proposition 12.

2) If $x_i^{(m)} \geq 0, x_i^{(m)} - y_i^{(m)} \leq 0$, then $x_i^{(m)} \leq y_i^{(m)}$, $x_i^{(m)2} \leq x_i^{(m)} \cdot y_i^{(m)}$ and $-x_i^{(m)} y_i^{(m)} \leq -x_i^{(m)2}$.

3) If $x_i^{(m)} < 0, x_i^{(m)} - y_i^{(m)} \geq 0$, then $x_i^{(m)} \geq y_i^{(m)}$, $x_i^{(m)2} \leq x_i^{(m)} \cdot y_i^{(m)}$ and $-x_i^{(m)} y_i^{(m)} \leq -x_i^{(m)2}$.

4) If $x_i^{(m)} < 0, x_i^{(m)} - y_i^{(m)} \leq 0$, then $y_i^{(m)} - x_i^{(m)} \leq \bar{L}\sqrt{n_2}a_m$, $y_i^{(m)} x_i^{(m)} - x_i^{(m)2} \geq \bar{L}\sqrt{n_2}a_m x_i^{(m)}$ and $-y_i^{(m)} x_i^{(m)} \leq -x_i^{(m)2} - \bar{L}\sqrt{n_2}a_m x_i^{(m)} \leq -x_i^{(m)2} + \bar{L}M\sqrt{n_2}a_m$. The last inequality is by Proposition 12.

All these four cases yield equation 52. $\qquad \square$

**Proposition 19** *Under the assumptions of Theorem 7, we have*

$$\begin{aligned} \bar{f}(\theta^{(m+1)}, \bar{\lambda}) \leq & \bar{f}(\theta^{(m)}, \bar{\lambda}) - \eta^{(m)} \|\nabla \bar{f}(\theta^{(m)}, \bar{\lambda})\|_2^2 \\ & + \eta^{(m)} \bar{L}M\sqrt{n_2}a_m + \frac{1}{2}(\eta^{(m)})^2 \cdot N\bar{L}M, \end{aligned} \tag{53}$$

*where $M$ is a constant and $a_m$ is defined in Proposition 17.*

*Proof.* By Proposition 13,
$$f_i(X_i : \tilde{\theta}, \lambda) \leq f_i(X_i : \hat{\theta}, \lambda) + \nabla f_i(X_i : \hat{\theta}, \lambda)^T(\tilde{\theta} - \hat{\theta}) + \frac{1}{2}\bar{L}\|\tilde{\theta} - \hat{\theta}\|_2^2.$$

Therefore, we can sum it over the entire training set from $i = 1$ to $N$ to obtain
$$\bar{f}(\tilde{\theta}, \lambda) \leq \bar{f}(\hat{\theta}, \lambda) + \nabla \bar{f}(\hat{\theta}, \lambda)^T(\tilde{\theta} - \hat{\theta}) + \frac{N}{2}\bar{L}\|\tilde{\theta} - \hat{\theta}\|_2^2. \tag{54}$$

In Algorithm 1, we define the update of $\theta$ in the following full gradient way:

$$\theta^{(m+1)} := \theta^{(m)} - \eta^{(m)} \cdot \sum_{i=1}^{N} \cdot \nabla f_i(X_i : \theta^{(m)}, \lambda^{(m)}), \tag{55}$$

which implies

$$\theta^{(m+1)} - \theta^{(m)} = -\eta^{(m)} \cdot \nabla \bar{f}(\theta^{(m)}, \lambda^{(m)}). \tag{56}$$

By equation 56 we have $\tilde{\theta} - \hat{\theta} = \theta^{(m+1)} - \theta^{(m)} = -\eta^{(m)} \nabla \bar{f}(\theta^{(m)}, \lambda^{(m)})$. We now substitute $\tilde{\theta} := \theta^{(m+1)}$, $\hat{\theta} := \theta^{(m)}$ and $\lambda := \bar{\lambda}$ into equation 54 to obtain

$$\bar{f}(\theta^{(m+1)}, \bar{\lambda})$$
$$\leq \bar{f}(\theta^{(m)}, \bar{\lambda}) - \eta^{(m)} \nabla \bar{f}(\theta^{(m)}, \bar{\lambda})^T \nabla \bar{f}(\theta^{(m)}, \lambda^{(m)}) + (\eta^{(m)})^2 \cdot \frac{N \bar{L} M}{2}$$
$$\leq \bar{f}(\theta^{(m)}, \bar{\lambda}) - \eta^{(m)} \|\nabla \bar{f}(\theta^{(m)}, \bar{\lambda})\|_2^2 + \eta^{(m)} \bar{L} M \sqrt{n_2} a_m$$
$$+ \frac{1}{2}(\eta^{(m)})^2 \cdot N \bar{L} M. \tag{57}$$

The first inequality is by plugging equation 56 into equation 54, the second inequality comes from Proposition 12 and the third inequality comes from Proposition 18. □

## 6.4 PROOF OF THEOREM 11

Here we show Theorem 11 as the consequence of Theorem 7 and Lemmas 8, 9 and 10.

### 6.4.1 PROOF OF LEMMA 8

Here we show Lemma 8 as the consequence of Lemmas 20, 21 and 22.

**Lemma 20** $\sum_{m=1}^{\infty} \sum_{i=m}^{\infty} \sum_{n=1}^{i} \alpha^{(i)} \eta^{(n)} < \infty$ and $\sum_{m=1}^{\infty} \sum_{n=m}^{\infty} \alpha^{(n)} < \infty$ is a set of sufficient condition to ensure

$$\sum_{m=1}^{\infty} |\bar{\sigma}_j - \sigma_j^{(m)}| < \infty, \forall j. \tag{58}$$

*Proof.* By plugging equation 45 and equation 43 into equation 58, we have the following for all $j$:

$$\sum_{m=1}^{\infty} \left|\bar{\sigma}_j - \sigma_j^{(m)}\right| \leq \sum_{m=1}^{\infty} \left(|\tilde{\sigma}_j - \tilde{\sigma}_j^{(m)}| + \frac{\bar{\sigma}_j}{C} \sum_{n=m+1}^{\infty} \alpha^{(n)}\right)$$
$$\leq \frac{|k| \cdot \tilde{M}_{\bar{L}, M}}{C} \sum_{m=1}^{\infty} \sum_{i=m}^{\infty} \alpha^{(i)} \sum_{j=1}^{i} \eta^{(j)} + \frac{\bar{\sigma}_j + |k||\bar{C}|}{C} \sum_{m=1}^{\infty} \sum_{n=m+1}^{\infty} \alpha^{(n)}. \tag{59}$$

It is easy to see that the the following conditions are sufficient for right-hand side of equation 59 to be finite: $\sum_{m=1}^{\infty} \sum_{i=m}^{\infty} \sum_{n=1}^{i} \alpha^{(i)} \eta^{(n)} < \infty$ and $\sum_{m=1}^{\infty} \sum_{n=m}^{\infty} \alpha^{(n)} < \infty$.

Therefore, we obtain $\qquad \sum_{m=1}^{\infty} |\bar{\sigma}_j - \sigma_j^{(m)}| < \infty, \forall j.$ □

**Lemma 21** *Under Assumption 4,*

$$\sum_{m=1}^{\infty} \sum_{i=m}^{\infty} \sum_{n=1}^{i} \alpha^{(i)} \eta^{(n)} < \infty \quad and \quad \sum_{m=1}^{\infty} \sum_{n=m}^{\infty} \alpha^{(n)} < \infty$$

*is a set of sufficient conditions to ensure*

$$\limsup_{M \to \infty} \sum_{m=1}^{M} \left|\bar{f}(\theta^{(m)}, \lambda^{(m)}) - \bar{f}(\theta^{(m)}, \bar{\lambda})\right| < \infty.$$

*Proof.* By Assumption 4, we have

$$\|l_i(x) - l_i(y)\| \leq \hat{M} \|x - y\| \leq \hat{M} \sum_{i=1}^{D} |x_i - y_i|. \tag{60}$$

By the definition of $f_i(\cdot)$, we then have

$$\sum_{m=1}^{\infty} \left| \bar{f}(\theta^{(m)}, \lambda^{(m)}) - \bar{f}(\theta^{(m)}, \bar{\lambda}) \right| \tag{61}$$

$$\leq \sum_{m=1}^{\infty} \sum_{i=1}^{N} \left| \left( l_i(X_i : \theta^{(m)}, \lambda^{(m)}) - l_i(X_i : \theta^{(m)}, \bar{\lambda}) \right) \right| \tag{62}$$

$$\leq M_2 \sum_{m=1}^{\infty} \sum_{j=1}^{D} \sum_{i=1}^{N} \left| \frac{a(W_{1,j,\cdot}^{(m)} X_i) - \mu_j^{(m)}}{\sigma_j^{(m)} + \epsilon_B} - \frac{a(W_{1,j,\cdot}^{(m)} X_i) - \bar{\mu}_j}{\bar{\sigma}_j + \epsilon_B} \right| \tag{63}$$

$$\leq M_3 \sum_{m=1}^{\infty} \sum_{j=1}^{D} \left( \sum_{i=1}^{N} |k| |W_{1,j,\cdot}^{(m)} X_i| \left| \frac{\bar{\sigma}_j - \sigma_j^{(m)}}{\epsilon_B^2} \right| + N \left| \frac{\bar{\mu}_j}{\bar{\sigma}_j + \epsilon_B} - \frac{\mu_j^{(m)}}{\sigma_j^{(m)} + \epsilon_B} \right| \right). \tag{64}$$

The first inequality is by the Cauchy-Schwarz inequality, and the second one is by equation 60. To show the finiteness of equation 64, we only need to show the following two statements:

$$\sum_{m=1}^{\infty} \sum_{i=1}^{N} |k| |W_{1,j,\cdot}^{(m)} X_i| \left| \frac{\bar{\sigma}_j - \sigma_j^{(m)}}{\epsilon_B^2} \right| < \infty, \forall j \tag{65}$$

and

$$\sum_{m=1}^{\infty} \left| \frac{\bar{\mu}_j}{\bar{\sigma}_j + \epsilon_B} - \frac{\mu_j^{(m)}}{\sigma_j^{(m)} + \epsilon_B} \right| < \infty, \forall j. \tag{66}$$

*Proof of equation 65:* For all $j$ we have

$$\sum_{m=1}^{\infty} \sum_{i=1}^{N} |k| |W_{1,j,\cdot}^{(m)} X_i| \left| \frac{\bar{\sigma}_j - \sigma_j^{(m)}}{\epsilon_B^2} \right|$$

$$\leq \sum_{m=1}^{\infty} |k| N D M \max_i \|X_i\| \frac{1}{\epsilon_B^2} \left| \bar{\sigma}_j - \sigma_j^{(m)} \right| \tag{67}$$

$$= |k| N D M \max_i \|X_i\| \frac{1}{\epsilon_B^2} \sum_{m=1}^{\infty} \left| \bar{\sigma}_j - \sigma_j^{(m)} \right|.$$

The inequality comes from $|W_{1,j,\cdot}^{(m)} X_i| \leq D M \|X_i\|_2$, where $D$ is the dimension of $X_i$ and $M$ is the element-wise upper bound for $W_{1,j,\cdot}^{(m)}$ in Assumption 2.

Finally, we invoke Lemma 14 to assert that $\sum_{m=1}^{\infty} \left| \bar{\sigma}_j - \sigma_j^{(m)} \right|$ is finite.

*Proof of equation 66:* For all $j$ we have

$$\sum_{m=1}^{\infty} \left| \frac{\bar{\mu}_j}{\bar{\sigma}_j + \epsilon_B} - \frac{\mu_j^{(m)}}{\sigma_j^{(m)} + \epsilon_B} \right|$$
$$\leq \sum_{m=1}^{\infty} \left| \frac{\bar{\mu}_j}{\bar{\sigma}_j + \epsilon_B} - \frac{\mu_j^{(m)}}{\bar{\sigma}_j + \epsilon_B} \right| + \sum_{m=1}^{\infty} \left| \frac{\mu_j^{(m)}}{\bar{\sigma}_j + \epsilon_B} - \frac{\mu_j^{(m)}}{\sigma_j^{(m)} + \epsilon_B} \right|. \tag{68}$$

The first term in equation 68 is finite since $\{\mu_j^{(m)}\}$ is a Cauchy series. For the second term, we know that there exists a constant $M$ such that for all $m \geq M$, $\mu_j^{(m)} \leq \bar{\mu} + 1$. This is also by the fact that $\{\mu_j^{(m)}\}$ is a Cauchy series and it converges to $\bar{\mu}$. Therefore, the second term in equation 68 becomes

$$\sum_{m=1}^{M-1} \left| \frac{\mu_j^{(m)}}{\bar{\sigma}_j + \epsilon_B} - \frac{\mu_j^{(m)}}{\sigma_j^{(m)} + \epsilon_B} \right| + \sum_{m=M}^{\infty} \left| \frac{\mu_j^{(m)}}{\bar{\sigma}_j + \epsilon_B} - \frac{\mu_j^{(m)}}{\sigma_j^{(m)} + \epsilon_B} \right|$$

$$\leq \sum_{m=1}^{M-1} \left| \frac{\mu_j^{(m)}}{\bar{\sigma}_j + \epsilon_B} - \frac{\mu_j^{(m)}}{\sigma_j^{(m)} + \epsilon_B} \right| + \sum_{m=M}^{\infty} (\bar{\mu} + 1) \left| \frac{1}{\bar{\sigma}_j + \epsilon_B} - \frac{1}{\sigma_j^{(m)} + \epsilon_B} \right|. \tag{69}$$

Noted that function $f(\sigma) = \dfrac{1}{\sigma + \epsilon_B}$ is Lipschitz continuous since its gradient is bounded by $\dfrac{1}{\epsilon_B^2}$. Therefore we can choose $\dfrac{1}{\epsilon_B^2}$ as the Lipschitz constant for $f(\sigma)$. We then have the following inequality:

$$\left| \frac{1}{\bar{\sigma}_j + \epsilon_B} - \frac{1}{\sigma_j^{(m)} + \epsilon_B} \right| \leq \frac{1}{\epsilon_B^2} |\bar{\sigma}_j - \sigma_j^{(m)}|. \tag{70}$$

Plugging equation 70 into equation 69, we obtain

$$
\begin{aligned}
&\sum_{m=1}^{M-1} \left| \frac{\mu_j^{(m)}}{\bar{\sigma}_j + \epsilon_B} - \frac{\mu_j^{(m)}}{\sigma_j^{(m)} + \epsilon_B} \right| + \sum_{m=M}^{\infty} (\bar{\mu} + 1) \left| \frac{1}{\bar{\sigma}_j + \epsilon_B} - \frac{1}{\sigma_j^{(m)} + \epsilon_B} \right| \\
&\leq \sum_{m=1}^{M-1} \left| \frac{\mu_j^{(m)}}{\bar{\sigma}_j + \epsilon_B} - \frac{\mu_j^{(m)}}{\sigma_j^{(m)} + \epsilon_B} \right| + \sum_{m=M}^{\infty} \frac{(\bar{\mu} + 1)}{\epsilon_B^2} |\bar{\sigma}_j - \sigma_j^{(m)}|,
\end{aligned}
\tag{71}
$$

where the first term is finite by the fact that $M$ is a finite constant. We have shown the condition for the second term to be finite in Lemma 20. Therefore,

$$\sum_{m=1}^{\infty} \left| \frac{\bar{\mu}_j}{\bar{\sigma}_j + \epsilon_B} - \frac{\mu_j^{(m)}}{\sigma_j^{(m)} + \epsilon_B} \right| < \infty, \forall j.$$

By equation 65 and equation 66, we have that the right-hand side of equation 64 is finite. It means that the left-hand side of equation 64 is finite. Thus,

$$\sum_{m=1}^{\infty} \left| \bar{f}(\theta^{(m)}, \lambda^{(m)}) - \bar{f}(\theta^{(m)}, \bar{\lambda}) \right| < \infty. \qquad \square$$

**Lemma 22** *If*

$$\sum_{m=1}^{\infty} \sum_{i=m}^{\infty} \sum_{n=1}^{i} \alpha^{(i)} \eta^{(n)} < \infty \quad and \quad \sum_{m=1}^{\infty} \sum_{n=m}^{\infty} \alpha^{(n)} < \infty,$$

*then*

$$\limsup_{M \to \infty} \sum_{m=1}^{M} \eta^{(m)} \|\nabla \bar{f}(\theta^{(m)}, \bar{\lambda})\|_2^2 < \infty.$$

*Proof.* For simplicity of the proof, we define

$$
\begin{aligned}
T^{(M)} &:= \sum_{m=1}^{M} \eta^{(m)} \|\nabla \bar{f}(\theta^{(m)}, \bar{\lambda})\|_2^2, \\
O^{(m)} &:= \bar{f}(\theta^{(m+1)}, \lambda^{(m+1)}) - \bar{f}(\theta^{(m)}, \lambda^{(m)}), \\
\Delta_1^{(m+1)} &:= \bar{f}(\theta^{(m+1)}, \lambda^{(m+1)}) - \bar{f}(\theta^{(m+1)}, \bar{\lambda}), \\
\Delta_2^{(m)} &:= \bar{f}(\theta^{(m+1)}, \bar{\lambda}) - \bar{f}(\theta^{(m)}, \bar{\lambda}),
\end{aligned}
$$

where $\bar{\lambda}$ is the converged value of $\lambda$ in Theorem 7. Therefore,

$$O^{(m)} = \Delta_1^{(m+1)} + \Delta_1^{(m)} + \Delta_2^{(m)} \leq |\Delta_1^{(m+1)}| + |\Delta_1^{(m)}| + \Delta_2^{(m)}. \tag{72}$$

By Proposition 19,

$$\Delta_2^{(m)} \leq -\eta^{(m)} \|\nabla \bar{f}(\theta^{(m)}, \bar{\lambda})\|_2^2 + \eta^{(m)} \bar{L} M \sqrt{n_2} a_m + \frac{1}{2} (\eta^{(m)})^2 \cdot N \bar{L} M. \tag{73}$$

We sum the inequality equation 72 from 1 to $K$ with respect to $m$ and plug equation 73 into it to obtain

$$
\begin{aligned}
\sum_{m=1}^{K} O^{(m)} &\leq \sum_{m=1}^{K} |\Delta_1^{(m+1)}| + \sum_{m=1}^{K} |\Delta_1^{(m)}| - \sum_{m=1}^{K} \{\eta^{(m)} \|\nabla \bar{f}(\theta^{(m)}, \bar{\lambda})\|_2^2\} \\
&\quad + \sum_{m=1}^{K} \eta^{(m)} \bar{L} M \sqrt{n_2} a_m + \sum_{m=1}^{K} \{\frac{1}{2} (\eta^{(m)})^2 N \bar{L} M\} \\
&= \sum_{m=1}^{K} |\Delta_1^{(m+1)}| + \sum_{m=1}^{K} |\Delta_1^{(m)}| - T^{(K)} \\
&\quad + \bar{L}^2 \sqrt{n_2} \cdot \sum_{m=1}^{K} \eta^{(m)} a_m + \sum_{m=1}^{K} \{\frac{1}{2} (\eta^{(m)})^2 N \bar{L} M\}.
\end{aligned}
\tag{74}
$$

From this, we have:

$$
\begin{aligned}
\limsup_{K\to\infty} T^{(K)} \leq {}& \limsup_{K\to\infty} \frac{-1}{c_1}(\bar{f}(\theta^{(K)}, \lambda^{(K)}) - \bar{f}(\theta^{(1)}, \lambda^{(1)})) \\
& + \limsup_{K\to\infty} \frac{1}{c_1} \sum_{m=1}^{K} (|\Delta_1^{(m+1)}| + |\Delta_1^{(m)}|) \\
& + \limsup_{K\to\infty} \bar{L}^2 \sqrt{n_2} \sum_{m=1}^{K} \eta^{(m)} a_m \\
& + \limsup_{K\to\infty} \frac{N\bar{L}K}{2c_1} \sum_{m=1}^{K} \eta^{(m)2}.
\end{aligned}
\tag{75}
$$

Next we show that each of the four terms in the right-hand side of equation 75 is finite, respectively. For the first term,

$$
\limsup_{K\to\infty} \frac{-1}{c_1}(\bar{f}(\theta^{(K)}, \lambda^{(K)}) - \bar{f}(\theta^{(1)}, \lambda^{(1)})) < \infty
\tag{76}
$$

is by the fact that the parameters $\{\theta, \lambda\}$ are in compact sets, which implies that the image of $f_i(\cdot)$ is in a bounded set.

For the second term, we showed its finiteness in Lemma 21.

For the third term, by equation 42, we have

$$
\begin{aligned}
& \limsup_{K\to\infty} \sum_{m=1}^{K} \eta^{(m)} a_m \\
={}& \limsup_{K\to\infty} \sum_{m=1}^{K} \eta^{(m)} \left( K_1 \sum_{i=m}^{\infty} \sum_{j=1}^{i} \alpha^{(i)} \eta^{(j)} + K_2 \sum_{i=m}^{\infty} \alpha^{(i)} \right) \\
={}& K_1 \limsup_{K\to\infty} \sum_{m=1}^{K} \eta^{(m)} \left( \sum_{i=m}^{\infty} \sum_{j=1}^{i} \alpha^{(i)} \eta^{(j)} \right) + K_2 \limsup_{K\to\infty} \sum_{m=1}^{K} \eta^{(m)} \sum_{i=m}^{\infty} \alpha^{(i)}.
\end{aligned}
\tag{77}
$$

The right-hand side of equation 77 is finite because

$$
\sum_{m=1}^{\infty} \eta^{(m)} \left( \sum_{i=m}^{\infty} \sum_{j=1}^{i} \alpha^{(i)} \eta^{(j)} \right) < \sum_{m=1}^{\infty} \left( \sum_{i=m}^{\infty} \sum_{j=1}^{i} \alpha^{(i)} \eta^{(j)} \right) < \infty
\tag{78}
$$

and

$$
\sum_{m=1}^{\infty} \eta^{(m)} \sum_{i=m}^{\infty} \alpha^{(i)} < \sum_{m=1}^{\infty} \sum_{i=m}^{\infty} \alpha^{(i)} < \infty.
\tag{79}
$$

The second inequalities in equation 78 and equation 79 come from the stated assumptions of this lemma.

For the fourth term,

$$
\limsup_{K\to\infty} \frac{N\bar{L}M}{2c} \sum_{m=1}^{K} \eta^{(m)2} < \infty
\tag{80}
$$

holds, because we have $\sum_{m=1}^{\infty} (\eta^{(m)})^2 < \infty$ in Assumption 3. Therefore, $T^{(\infty)} = \sum_{m=1}^{\infty} \eta^{(m)} \|\nabla \bar{f}(\theta^{(m)}, \bar{\lambda})\|_2^2 < \infty$ holds. $\qquad\square$

In Lemmas 20, 21 and 22, we show that $\{\sigma^{(m)}\}$ and $\{\mu^{(m)}\}$ are Cauchy series, hence Lemma 8 holds.

### 6.4.2 PROOF OF LEMMA 9

This proof is similar to the the proof by Bertsekas & Tsitsiklis (2000).

*Proof.* By Theorem 8, we have

$$
\limsup_{M\to\infty} \sum_{m=1}^{M} \eta^{(m)} \|\nabla \bar{f}(\theta^{(m)}, \bar{\lambda})\|_2^2 < \infty.
\tag{81}
$$

If there exists a $\epsilon > 0$ and an integer $\bar{m}$ such that

$$
\|\nabla \bar{f}(\theta^{(m)}, \bar{\lambda})\|_2 \geq \epsilon
$$

for all $m \geq \bar{m}$, we would have

$$\liminf_{M \to \infty} \sum_{m=\bar{m}}^{M} \eta^{(m)} \|\nabla \bar{f}(\theta^{(m)}, \bar{\lambda})\|_2^2 \geq \liminf_{M \to \infty} \epsilon^2 \sum_{m=\bar{m}}^{M} \eta^{(m)} = \infty \tag{82}$$

which contradicts equation 81. Therefore, $\liminf\limits_{m \to \infty} \|\nabla \bar{f}(\theta^{(m)}, \bar{\lambda})\|_2 = 0$. □

### 6.4.3 PROOF OF LEMMA 10

**Lemma 23** *Let $Y_t, W, t$ and $Z_t$ be three sequences such that $W_t$ is nonnegative for all $t$. Assume that*

$$Y_{t+1} \leq Y_t - W_t + Z_t, \quad t = 0, 1, ..., \tag{83}$$

*and that the series $\sum_{t=0}^{T} Z_t$ converges as $T \to \infty$. Then either $Y_t \to \infty$ or else $Y_t$ converges to a finite value and $\sum_{t=0}^{\infty} W_t < \infty$.*

This lemma has been proven by Bertsekas & Tsitsiklis (2000).

**Lemma 24** *When*

$$\sum_{m=1}^{\infty} \sum_{i=m}^{\infty} \sum_{n=1}^{i} \alpha^{(i)} \eta^{(n)} < \infty \quad \text{and} \quad \sum_{m=1}^{\infty} \sum_{n=m}^{\infty} \alpha^{(n)} < \infty, \tag{84}$$

*it follows that $\bar{f}(\theta^{(m)}, \bar{\lambda})$ converge to a finite value.*

*Proof.* By Proposition 19, we have

$$\begin{aligned} \bar{f}(\theta^{(m+1)}, \bar{\lambda}) \leq & \bar{f}(\theta^{(m)}, \bar{\lambda}) - \eta^{(m)} \|\nabla \bar{f}(\theta^{(m)}, \bar{\lambda})\|_2^2 \\ & + \eta^{(m)} \bar{L} M \sqrt{n_2} a_m + \frac{1}{2} (\eta^{(m)})^2 \cdot N \bar{L} M. \end{aligned} \tag{85}$$

Let $Y^{(m)} := \bar{f}(\theta^{(m)}, \bar{\lambda})$, $W^{(m)} := \eta^{(m)} \|\nabla \bar{f}(\theta^{(m)}, \bar{\lambda})\|_2^2$ and $Z^{(m)} := \eta^{(m)} \bar{L} M \sqrt{n_2} a_m + \frac{1}{2} (\eta^{(m)})^2 \cdot N \bar{L} M$. By equation 10 and equation 77- equation 79, it is easy to see that $\sum_{m=0}^{M} Z^{(m)}$ converges as $M \to \infty$. Therefore, by Lemma 23, $Y^{(m)}$ converges to a finite value. The infinite case can not occur in our setting due to Assumptions 1 and 2. □

**Lemma 25** *If*

$\sum_{m=1}^{\infty} \sum_{i=m}^{\infty} \sum_{n=1}^{i} \alpha^{(i)} \eta^{(n)} < \infty \quad \text{and} \quad \sum_{m=1}^{\infty} \sum_{n=m}^{\infty} \alpha^{(n)} < \infty,$

*then* $\lim\limits_{m \to \infty} \|\nabla \bar{f}(\theta^{(m)}, \bar{\lambda})\|_2 = 0$.

*Proof.* To show that $\lim\limits_{m \to \infty} \|\nabla \bar{f}(\theta^{(m)}, \bar{\lambda})\|_2 = 0$, assume the contrary; that is,

$$\limsup_{m \to \infty} \|\nabla \bar{f}(\theta^{(m)}, \bar{\lambda})\|_2 > 0.$$

Then there exists an $\epsilon > 0$ such that $\|\nabla \bar{f}(\theta^{(m)}, \bar{\lambda})\| < \epsilon/2$ for infinitely many $m$ and also $\|\nabla \bar{f}(\theta^{(m)}, \bar{\lambda})\| > \epsilon$ for infinitely many $m$. Therefore, there is an infinite subset of integers $\mathbb{M}$, such that for each $m \in \mathbb{M}$, there exists an integer $q(m) > m$ such that

$$\begin{aligned} & \|\nabla \bar{f}(\theta^{(m)}, \bar{\lambda})\| < \epsilon/2, \\ & \|\nabla \bar{f}(\theta^{(i(m))}, \bar{\lambda})\| > \epsilon, \\ & \epsilon/2 \leq \|\nabla \bar{f}(\theta^{(i)}, \bar{\lambda})\| \leq \epsilon, \\ & \quad \text{if } m < i < q(m). \end{aligned} \tag{86}$$

From $\|\nabla \bar{f}(\theta^{(m+1)}, \bar{\lambda})\| - \|\nabla \bar{f}(\theta^{(m)}, \bar{\lambda})\| \leq \bar{L} \eta^{(m)} \|\nabla \bar{f}(\theta^{(m)}, \lambda^{(m)})\|$, it follows that for all $m \in \mathbb{M}$ that are sufficiently large so that $\bar{L} \eta^{(m)} < \epsilon/4$, we have

$$\epsilon/4 \leq \|\nabla \bar{f}(\theta^{(m)}, \lambda^{(m)})\|. \tag{87}$$

Otherwise the condition $\epsilon/2 \leq \|\nabla \bar{f}(\theta^{(m+1)}, \bar{\lambda})\|$ would be violated. Without loss of generality, we assume that the above relations as well as equation 57 hold for all $m \in \mathbb{M}$. With the above observations, we have for all $m \in \mathbb{M}$,

$$
\begin{aligned}
\frac{\epsilon}{2} &\leq \|\nabla \bar{f}(\theta^{q(m)}, \bar{\lambda})\| - \|\nabla \bar{f}(\theta^{(m)}, \bar{\lambda})\| \leq \bar{L}\|\theta^{q(m)} - \theta^{(m)}\| \\
&\leq \bar{L} \sum_{i=m}^{q(m)-1} \eta^{(i)}(\|\nabla \bar{f}(\theta^{(i)}, \bar{\lambda})\| + \|\nabla \bar{f}(\theta^{(i)}, \lambda^{(i)}) - \nabla \bar{f}(\theta^{(i)}, \bar{\lambda})\|) \\
&= \bar{L}\epsilon \sum_{i=m}^{q(m)-1} \eta^{(i)} + \bar{L}^2 \sqrt{n_2} M_1 \sum_{i=m}^{q(m)-1} \eta^{(i)} \sum_{j=m}^{\infty} \sum_{k=1}^{j} \alpha^{(j)} \eta^{(k)} \\
&\quad + \bar{L}^2 \sqrt{n_2} M_2 \sum_{i=m}^{q(m)-1} \eta^{(i)} \sum_{j=m}^{\infty} \alpha^{(j)}
\end{aligned}
\tag{88}
$$

The first inequality is by equation 86 and the third one is by the Lipschitz condition assumption. The seventh one is by equation 51. By equation 12, we have for all $m \in \mathbb{M}$,

$$
\sum_{i=m}^{q(m)-1} \eta^{(i)} \sum_{j=m}^{\infty} \sum_{k=1}^{j} \alpha^{(j)} \eta^{(k)} < \sum_{i=1}^{\infty} \sum_{j=i}^{\infty} \sum_{k=1}^{j} \alpha^{(j)} \eta^{(k)} < \infty
\tag{89}
$$

and

$$
\sum_{i=m}^{q(m)-1} \eta^{(i)} \sum_{j=m}^{\infty} \alpha^{(j)} < \sum_{i=1}^{\infty} \sum_{j=i}^{\infty} \alpha^{(j)} < \infty.
\tag{90}
$$

It is easy to see that for any sequence $\{\alpha_i\}$ with $\sum_{i=1}^{\infty} \alpha_i < \infty$, if follows that $\liminf_{M \to \infty} \sum_{i=M}^{\infty} \alpha_i = 0$. Therefore, $\liminf_{m \to \infty} \sum_{i=m}^{q(m)-1} \eta^{(i)} \sum_{j=m}^{\infty} \sum_{k=1}^{j} \alpha^{(j)} \eta^{(k)} = 0$ and $\liminf_{m \to \infty} \sum_{i=m}^{q(m)-1} \eta^{(i)} \sum_{j=m}^{\infty} \alpha^{(j)} = 0$. From this it follows that

$$
\liminf_{m \to \infty} \sum_{i=m}^{q(m)-1} \eta^{(i)} \geq \frac{1}{2\bar{L}}.
\tag{91}
$$

By equation 51 and equation 87, if we pick $m \in \mathbb{M}$ such that $L\sqrt{n_2} a_m \leq \frac{\epsilon}{8}$, we have $\|\nabla \bar{f}(\theta^{(m)}, \bar{\lambda})\| \geq \frac{\epsilon}{8}$. Using equation 57, we observe that

$$
\begin{aligned}
\bar{f}(\theta^{q(m)}, \bar{\lambda}) \\
\leq \bar{f}(\theta^{(m)}, \bar{\lambda}) - c_1 \left(\frac{\epsilon}{8}\right)^2 \sum_{i=m}^{q(m)-1} \eta^{(i)} + \frac{1}{2} \cdot N\bar{L}M \sum_{i=m}^{q(m)-1} (\eta^{(i)})^2, \forall m \in \mathbb{M},
\end{aligned}
\tag{92}
$$

where the second inequality is by equation 87. By Lemma 24, $\bar{f}(\theta^{q(m)}, \bar{\lambda})$ and $\bar{f}(\theta^{(m)}, \bar{\lambda})$ converge to the same finite value. Using this convergence result and the assumption $\sum_{m=0}^{\infty} (\eta^{(m)})^2 < \infty$, this relation implies that

$$
\limsup_{m \to \infty, m \in \mathbb{M}} \sum_{i=m}^{q(m)-1} \eta^{(i)} = 0 \text{ and contradicts equation 91.} \qquad \square
$$

By Lemmas 23, 24 and 25, we show that Theorem 11 holds.

## 6.5 DISCUSSIONS OF CONDITIONS FOR STEPSIZES

Here we discuss the actual conditions for $\eta^{(m)}$ and $\alpha^{(m)}$ to satisfy the assumptions of Theorem 7 and Lemma 8. We only consider the cases $\eta^{(m)} = \frac{1}{m^k}$ and $\alpha^{(m)} = \frac{1}{m^h}$, but the same analysis applies to the cases $\eta^{(m)} = O(\frac{1}{m^k})$ and $\alpha^{(m)} = O(\frac{1}{m^h})$.

## 6.6 ASSUMPTIONS OF THEOREM 7

For the assumptions of Theorem 7, the first condition $\sum_{m=1}^{\infty} \alpha^{(m)} < \infty$ requires $h > 1$. Besides, the second condition

$$
\sum_{m=1}^{\infty} \sum_{n=1}^{m} \alpha^{(m)} \eta^{(n)} \approx \frac{1}{h-1} \sum_{n=1}^{\infty} \eta^{(n)} \frac{1}{n^{h-1}} = \frac{1}{h-1} \sum_{n=1}^{\infty} \frac{1}{n^{k+h-1}} < \infty
\tag{93}
$$

requires $k + h > 2$. The approximation comes from the fact that for every $p > 1$, we have

$$
\sum_{k=n}^{\infty} k^{-p} \approx \int_{k=n}^{\infty} k^{-p} dx = \frac{1}{1-p} x^{1-p} \Big|_{n}^{\infty} = \frac{1}{p-1} \frac{1}{n^{p-1}}.
\tag{94}
$$

Since $k \geq 1$ due to Assumption 3, we conclude that $k + h > 2$. Therefore, the conditions for $\eta^{(m)}$ and $\alpha^{(m)}$ to satisfy the assumptions of Theorem 7 are $h > 1$ and $k \geq 1$.

## 6.7 ASSUMPTIONS OF LEMMA 8

For the assumptions of Theorem 7, the first condition

$$\sum_{m=1}^{\infty} \sum_{n=m}^{\infty} \alpha^{(n)} \approx \sum_{m=1}^{\infty} \frac{1}{m^{h-1}} < \infty \tag{95}$$

requires $h > 2$.

Besides, the second condition is

$$\sum_{m=1}^{\infty} \sum_{i=m}^{\infty} \sum_{n=1}^{i} \alpha^{(i)} \eta^{(n)} = \sum_{m=1}^{\infty} \sum_{i=m}^{\infty} \alpha^{(i)} \sum_{n=1}^{i} \eta^{(n)} \leq C \sum_{m=1}^{\infty} \sum_{i=m}^{\infty} \alpha^{(i)} < \infty. \tag{96}$$

The inequality holds because for any $p > 1$, we have

$$\sum_{k=1}^{n} k^{-p} \approx \int_{k=1}^{n} k^{-p} dk = \frac{1}{1-p} k^{1-p} \Big|_{1}^{n} = \frac{1}{p-1}(1 - n^{1-p}) \leq C \tag{97}$$

Therefore, the conditions for $\eta^{(m)}$ and $\alpha^{(m)}$ to satisfy the assumptions of Lemma 8 are $h > 2$ and $k \geq 1$.

