# OpenReview forum: "Diminishing Batch Normalization"
_ICLR.cc/2019/Conference_

### Official Review · AnonReviewer2 · 2018-10-31
**Review: Diminishing Batch normalization.**

**Rating:** 4
**Confidence:** 5

**Review:**


In this work, the authors propose a generalization of the batch normalization (BN) technique often used in training neural networks, and analyzed this convergence. In particular, a one hidden layer and one BN hidden layer fully connected network is considered, and a deterministic gradient descent algorithm with certain kind of BN has been considered in this work. The proposed “generalized” BN strategy is devised on the deterministic setting, but it is a slight generalization of the original BN by introducing a moving average operation. Classical results of Bertsekas is leveraged to show the asymptotic convergence of the algorithm.

I have the following three main comments about the paper.
1)	Only deterministic setting is considered, but in this case every time the entire data set will be used to perform the averaging, it appears to be much easier to analyze than the stochastic setting. Further the reviewer has doubt on whether the resulting deterministic algorithm has any practice value.
2)	Because the authors have used the Bertsekas/Tsitsiklis (B/T) argument, only asymptotic convergence is shown. It is not clear, even in the deterministic case, whether some kind of sublinear convergence rate can be obtained.
3)	Only one hidden layer of neural network with one BN operation is considered. It is not clear whether the analysis can be extended to multiple layers, despite the statement of the author saying that “the technique presented can be extended to more layers with additional notation”. In particular, when there are multiple layers, the BN layers will be further composite together across multiple nonlinear operations.
4)	The authors have mentioned that the derivative is always taking w.r.t. theta. However, in (9) is appears that the derivative is taken with respect to lambda, in order to get the Lipschitz condition on \lambda. This is a bit confusing. Also it is not clear how the gradient in Assumption 5 is defined.
5)	Assumption 5 does not make sense. Problem (1) is a constrained problem with both variables being confined in compact feasible sets. And this condition is important in Assumption2. Now the authors say that at stationary solution the gradient is zero? Please specify functions when this will happen. I will suggest that the authors use a proper definition of stationarity solution for constrained problems.
6)	Follow up on the previous point. The analysis builds upon B/T argument for unconstrained optimization. However it is not suitable for the constrained problems that the authors started out at the beginning of the paper. The authors may consider develop new analysis tools to understand the problem at hand, rather than assuming away the difficulties.

---

> ### Author Response · Authors · 2018-11-14
> **Rebuttal**
>
> Thanks for your delightful review. Please allow us to try to address your remarks below:
>
> 1) We believe it is possible to extend the current analysis to a mini-batch setting by reusing most of the present analysis. We point out that in numerical experiments we use mini-batches.
>
> 2) Our analysis does not draw any conclusion regarding the convergence rate. This is an exciting future direction.
>
> 3) We agree that having more layers makes the problem more complicated. However, the analysis would essentially be the same except that the notation would be substantially more complex (but the proof ideas remain the same).
>
> 4) Although the function in (9) takes lambda as part of the input, the derivative notation we use here is only with respect to theta (and not including lambda). We actually note in the line before (2) that all of the derivatives in this work are taken with respect to theta not lambda. We should claim this again at (9) as a clarification.
>
> 5) We are not showing that the algorithm is converging to a local minimum but a point with the norm of gradient being zero. This is by definition a stationary point. It is unclear what happens with functions that do not satisfy Assumption 2 (bounded parameters) and 5 (stationary point existence).
>
> 6) Our analysis requires the parameters to be bounded within a compact set instead of any arbitrary constraints on parameters. Bounded parameters are required in the analysis to bound their norms. In practice, clipping is performed which is equivalent. Even in much simpler cases the proofs in an unbounded case are very complicated. The vast majority of convergence proofs in optimization assume bounded parameters.

---

### Official Review · AnonReviewer3 · 2018-11-03
**A momentum based approach for batch normalization with asymptotic convergence analysis**

**Rating:** 3
**Confidence:** 3

**Review:**

The authors propose a momentum based approach for batch normalization and provide an asymptotic convergence analysis of the objective in terms of the first order criterion. To my understanding, the main effort in the analysis is to show that the sequences of interest are Cauchy. Some numerical results are reported to demonstrate that the proposed variant of BN slightly outperforms BN with careful adjustment of some hyper parameter. The proposed approach is incremental, and the theoretical results are somewhat weak.

The most important issue is that the zero gradient of the objective function does not imply that it attains an (even local) minimum point. As for the 2-layer case, the objective function can be nonconvex in terms of the weight parameters with stationary points being saddle points, it is crucial to understand whether an iterative algorithm (GD or SGD) converges to a minimum point rather a saddle point. Thus, the first order criterion alone is not enough for this purpose, which is why extensive studies are carried out for nonconvex optimization (e.g., using both first and second order criteria for convergence [1]) and considering the specific structure of neural nets [2].

The analysis is somewhat confusing. The authors assume that the objective of interest have stationary points (\theta*, \lambda*), and also show that the sequence of the norm of gradient convergence to zero, with the \lambda^(m) converges to \bar{\lambda}. What is the relationship between \lambda* and \bar{\lambda}? It is not clear whether they are the same point or not. Moreover, since there is no converge of the parameter, it is not clear what the convergence for the \lambda imply here, as we also discussed above that the zero gradient itself may mean nothing.

In addition, the writing need improvements. Some statements are not accurate. For example, on page 3, after equation (2), the authors state “The deep network …”, though they mentioned it is for a 2-layer net. Also, more explicit explanation and definitions are necessary for notations. For example, it is clearer to define explicitly the parameters with \bar (e.g., for \lambda) as the limit point.

[1] Ge et al. Escaping from saddle points—online stochastic gradient for tensor decomposition.
[2] Li and Yuan. Convergence Analysis of Two-layer Neural Networks with ReLU Activation.

---

> ### Author Response · Authors · 2018-11-14
> **Rebuttal**
>
> Thanks for your interesting review. Please allow us to try to address your remarks below:
>
> We agree that our current analysis relies on the assumptions of bounded parameter space and a gradient zero point. The major contribution of our work is that we are the first to provide a convergence analysis when considering transformation with BN layers. We believe we never claim that we show convergence to a local minimum.
>
> We should have made this notation clearer in the paper. We have noted the difference in the analysis part. \lambda* stands for the lambda value at the stationary point, where the parameters are (\theta*, \lambda*). \bar{\lambda} is the value that our algorithm converges to in Theorem 7. We show in lambda 10 and Theorem 11 that this \bar{\lambda} eventually converges to \lambda*, where the loss function reaches zero gradients.
>
> We agree that having more layers makes the problem more complex. However, the analysis would essentially be the same except that the notation would be substantially more complex (but the proof ideas remain the same).

---

> > ### Comment · AnonReviewer3 · 2018-11-22
> > **Thanks for the update**
> >
> > The convergence to the stationary point rather than to the minimum is still the major concern for the strength/significance of the analysis for optimizing networks. I do not have further questions.

---

### Official Review · AnonReviewer1 · 2018-11-05
**interesting idea, empirical results are not convincing**

**Rating:** 4
**Confidence:** 4

**Review:**

The paper introduces a modification of batch normalization technique. Original
batch normalization normalizes minibatch examples using their mean and standard deviation.
The proposed version of batch normalization, called diminishing batch normalization, normalized
examples in the current minibatch using  mean and standard deviations that are weighted average
of mean and standard deviation from the current and all previous minibatches. The authors prove convergence of
batch gradient descent with diminished batch normalization. Also, the authors show empirically
that Adagrad optimization with diminishing batch normalization can find a better local minimum than
Adagrad optimization with original batch normalization.

The idea of diminishing normalization is very sound. However I was not convinced that it gives empirical advantage.
The paper says that Table 1 shows "the best result obtained from each choice of \alpha^m ". Probably the numbers in
this table were obtained using some particular choice of the number of epochs. Unfortunately I didn't find in the
paper any details about the choice of the number of epochs. If we choose the number of epochs that minimize validation
loss, then according to Figures 4(a) and 3(a), if \alpha^m=1 then the validation loss is minimized around epoch 55
and corresponding test error should be less than 2.2%. But the corresponding top left entry in Table 1 has error 2.7%.

Additional technical remarks:
1. The abstract says "we also show the sufficient and necessary conditions for the step sizes and diminishing weights to ensure the convergence". I didn't find necessary conditions in the paper.

2. The authors claim that they are not aware of any prior analysis of batch normalization. The papers at https://arxiv.org/abs/1805.11604 and
https://arxiv.org/abs/1806.02375 , published initially in 5-6/2018, provide interesting theoretical insights on batch normalization.

3. Sentence after equation (2): change from D_1 to D.

4. Usually batch normalization is applied before non-linear activation. According to equations 3-6, the paper applies
batch normalization after the nonlinear activation. My understanding is that convergence proof relies on the former architecture. Does section 5 use the former or the latter architecture?

5. I am not sure that fully connected neural network is an efficient architecture for MNIST dataset. I would like to
to see experiments with CNN and MNIST.

---

> ### Author Response · Authors · 2018-11-14
> **Rebuttal**
>
> Thanks for your delightful review. Please allow us to try to address your remarks below:
>
> We present the values in the table based on the best value on the validation dataset during the training period of 100 epochs. The training loss is standard and is minimized during training. This procedure is very standard in deep learning.
>
> 1) We have included the analysis for sufficient and necessary conditions in the appendix, which is also submitted. We can move the main result for these conditions from the appendix to the main paper.
>
> 2) These two works do not cover convergence analyses, and thus they are different from this work. Besides, these two works came out later than our initial release on arxiv (https://arxiv.org/abs/1705.08011) and therefore have been conducted after our work. We will definitely add these two citations to our paper.
>
> 3) We will change this typo.
>
> 4) There is a misunderstanding here. We always apply batch normalization before nonlinear activation. In equation (3) and everything that follows, we denote the nonlinear activation with α(⋅). The computation for batch normalized value is always after this α(⋅).
>
> 5) In this work, we are also showing the results with CNN on the Cifar-10 dataset, which is even a more complex dataset than MNIST. We can use CNN on MNIST, but the fact that the algorithm works on Cifar-10 with CNN is a great indication that the algorithm is suitable also for CNN.

---

### Meta-Review · Area_Chair1 · 2018-12-16
**A sound idea but no sufficient evidence of its effectiveness**

**Confidence:** 4
**Recommendation:** Reject

**Metareview:**

The paper introduces a modification of batch normalization technique. In contrast to the original batch normalization that normalizes minibatch examples using their mean and standard deviation, this modification uses weighted average
of mean and standard deviation from the current and all previous minibatches. The authors then provide some theoretical justification for the superiority of their variant of BatchNorm.

Unfortunately, the empirical demonstration of the improved performance seems not sufficient and thus fairly unconvincing.